# Eco-Friendly Synthesis of Water-Glass-Based Silica Aerogels via Catechol-Based Modifier

**DOI:** 10.3390/nano10122406

**Published:** 2020-12-01

**Authors:** Hyeonjung Kim, Kangyong Kim, Hyunhong Kim, Doo Jin Lee, Jongnam Park

**Affiliations:** 1School of Energy and Chemical Engineering, Ulsan National Institute of Science and Technology (UNIST), Ulsan 44919, Korea; khjguswnd1@unist.ac.kr (H.K.); cyanide@unist.ac.kr (K.K.); khh2008@unist.ac.kr (H.K.); 2DPG Project and Chemical Business Planning Team, SKC, Seoul 03142, Korea; doojin.lee@sk.com; 3Department of Biomedical Engineering, Ulsan National Institute of Science and Technology (UNIST), Ulsan 44919, Korea

**Keywords:** water-glass aerogel, catechol coating, eco-friendly, trimethylchlorosilane substitute, ambient pressure drying, thermal insulation

## Abstract

Silica aerogels have attracted much attention owing to their excellent thermal insulation properties. However, the conventional synthesis of silica aerogels involves the use of expensive and toxic alkoxide precursors and surface modifiers such as trimethylchlorosilane. In this study, cost-effective water-glass silica aerogels were synthesized using an eco-friendly catechol derivative surface modifier instead of trimethylchlorosilane. Polydopamine was introduced to increase adhesion to the SiO_2_ surface. The addition of 4-*tert*-butyl catechol and hexylamine imparted hydrophobicity to the surface and suppressed the polymerization of the polydopamine. After an ambient pressure drying process, catechol-modified aerogel exhibited a specific surface area of 377 m^2^/g and an average pore diameter of approximately 21 nm. To investigate their thermal conductivities, glass wool sheets were impregnated with catechol-modified aerogel. The thermal conductivity was 40.4 mWm^−1^K^−1^, which is lower than that of xerogel at 48.7 mWm^−1^K^−1^. Thus, by precisely controlling the catechol coating in the mesoporous framework, an eco-friendly synthetic method for aerogel preparation is proposed.

## 1. Introduction

Silica aerogels are highly mesoporous materials composed of interconnected silica backbones. Due to their porous nanostructures with air pockets, they have high specific surface areas (400–1000 m^2^g^−1^), low densities (<0.05 gcm^−3^), and low thermal conductivities (<20 mWm^−1^K^−1^) [1,2,3]. Thus, owing to their unique properties, aerogels have been widely applied in various industries as catalysts, absorbents, and heat-insulating materials [4,5,6,7,8,9,10]. However, conventional aerogel synthesis is difficult to commercialize because it requires expensive and toxic alkoxide precursors such as tetramethyl orthosilicate (TMOS) or tetraethyl orthosilicate (TEOS) for the sol–gel reaction and supercritical drying (SCD) processes, which require special instruments for high-pressure conditions to maintain their porous framework without shrinkage.

To solve these problems, researchers have developed wet gels using inexpensive sodium silicates, called water-glass, and applied the ambient pressure drying (APD) method instead of the supercritical process [11,12,13,14,15,16]. Minimization of the shrinkage of gels during APD requires a “spring-back” effect that implies repulsion throughout the framework through hydrophobization of the silica surface. Such hydrophobization can be achieved either by using co-precursors such as methyltrimethoxysilane (MTMS), phenyltriethoxysilane (PTES), or dimethyldiethoxysilane (DMDES), or by treating silylation agents such as trimethylchlorosilane (TMCS), but both methods are expensive and not environmentally friendly [17,18]. Therefore, the development of an inexpensive and eco-friendly surface treatment method is required, but research studies on alternatives of silylation agents on the surface of water-glass aerogels are limited. As a different approach for eco-friendly aerogel, many studies have been conducted to develop the aerogels based on organic networks with low toxicity such as cellulose, lignin, and chitosan-based aerogels and increase their mechanical properties [19,20,21,22,23]. However, these materials still have a disadvantage of relatively low flame retardancy than silica aerogel; it is necessary to find a proper surface modifier imparting the hydrophobicity to the inorganic silica aerogel without the silylation agents.

Catechol derivatives, which have a high affinity for metal oxide surfaces, have been studied as environmentally friendly surface modifiers. Das et al. found that a hydrophobic self-assembled monolayer (SAM) could be formed on silica using mussel-derived catechol molecules [24]. In addition, Wang et al. improved the mechanical strengths and textural properties of silica aerogels by using catechol molecules as co-precursors for gelation [25]. However, there is still a lack of research on the use of catechol molecules as hydrophobic surface coating agents for APD-processed aerogels.

In this work, we focused on applying the good adhesion properties of catechol derivatives to silica frameworks in order to form hydrophobic surfaces without toxic TMCS. Examples of catechol derivatives include 4-*tert*-butylcatechol (TBC) and polydopamine (PDA). The degree of polymerization and hydrophobicity can be controlled by alkylamine treatment, thereby optimizing the environment-friendly surface treatment method. Catechol surface-treated silica wet gel was dried at ambient pressure, and the presence and sizes of pores were compared with those of xerogel using a scanning electron microscope (SEM) and Barrett–Joyner–Halenda (BJH) analysis. In addition, the synthesized aerogel was impregnated into a glass wool sheet for the fabrication of a thermal insulation blanket. The thermal conductivity analysis results showed a high reduction in thermal conductivity for this blanket compared with that of the xerogel–glass wool composite. Thus, instead of toxic hydrophobic modifiers, catechol derivatives were used for the synthesis of APD-processed aerogels, which was found to be an eco-friendly and economical synthetic method.

## 2. Materials and Methods

### 2.1. Materials

Sodium silicate solution (water-glass, SiO_2_ content of 28–30 wt%, SiO_2_:Na_2_O = 3.52:1) was purchased from Young Il Chemical Co., Ltd., Incheon, Korea. Glass wool sheet was purchased from GF Chem., Pyeongtaek, Korea. TBC (≥99.0%), dopamine hydrochloride, hexylamine (HA, 99%), TMCS (≥98.0%), and hydrochloric acid (HCl, 37%) were purchased from Sigma-Aldrich, Seoul, Korea. Methanol, ethanol, n-hexane, and isopropanol were obtained from Samchun ChemicalsPohang, Korea. All chemicals were used without further purification.

### 2.2. Preparation of Water-Glass Based Silica Wet Gel and Aerogel-Impregnated Glass Wool Sheet

All steps of wet-gel preparation were conducted under ambient temperature and atmosphere. First, 60 mL of the water-glass solution was mixed with 350 mL of deionized water. Next, 10 mL of diluted water-glass solution was mixed with 3.5 mL of 10 vol% HCl solution and poured into a mold. The gelation process was completed within 1 h. Once the wet gel had formed, we added 5 mL of deionized water and then aged the gel at room temperature for 2 days to strengthen the gel network. After aging, the wet gel was pulverized into small pieces and washed 3 times with 20 mL of with deionized water before further surface modification. Aerogel-impregnated glass wool sheets were prepared by the following method. Prior to gelation and surface modification, glass wool sheets were cut to 4 cm × 6 cm × 1 cm cuboids_._ Subsequently, the glass wool sheets were soaked in 30 mL of diluted water-glass solution. Next, the gelation process was carried out by adding 10.5 mL of acidic HCl solution. Next the wet gel–glass wool composite was aged for 2 days. After aging, the composite was washed 3 times with deionized water before further surface modification. The hydrophobization and drying steps were performed in the same manner as the aerogel surface modification steps described below.

### 2.3. Preparation of TMCS-Modified Aerogel and Xerogel

For hydrophobization with TMCS, the wet gels formed in Section 2.2 were used. Firstly, the water in the wet gel was replaced with the isopropanol/hexane solution through several solvent exchange processes. After the solvent exchange process, the wet gels were chemically modified in 6.25 vol% TMCS/hexane solution for 24 h. The TMCS-modified wet gel was washed 3 times with hexane to remove residual modifiers and by-products. After the solvent was decanted, the hydrophobized wet gel was dried for 12 h at room temperature and ambient pressure; then, it was further dried in an oven at 60 °C for 12 h to obtain a TMCS-modified aerogel. The xerogel could be produced by the same drying process without surface modification.

### 2.4. Preparation of Catechol-Modified Aerogel

The wet gel prepared in Section 2.2 was used for the surface modification applying catechol. Then, the water in the wet gel was replaced with alcohol solvents such as methanol and ethanol through several solvent exchange processes. After the solvent exchange process, the wet gels were chemically modified in 10.5 mM catechol solution (TBC or a mixture of dopamine hydrochloride and TBC, 1:5 molar ratio) for 24 h. The 210 mM HA reagent was added to the catechol solution for hydrophobicity. The catechol-modified wet gel was washed 5 times with alcohol to remove residual modifiers and by-products. Finally, the surface-modified aerogel was dried for 12 h at room temperature and ambient pressure; then, it was further dried in an oven at 60 °C for 12 h.

### 2.5. Surface Coating of SiO_2_ Wafer

To demonstrate the properties of the catechol-coated surface, the coating on the silicon wafer was performed by a simple dip-coating method. We selected the silicon wafer with a 200 nm oxide layer as a substrate because its components are similar to that of silica aerogel surfaces, and the smoothness of the surface is helpful for elaborating film analysis. The coating solution contained 10.5 mM of dopamine hydrochloride (or a mixture of dopamine hydrochloride and TBC, 1:5 molar ratio) and 210 mM of HA in methanol (or ethanol). SiO_2_ substrates were immersed vertically in the coating solution for 12 h. The substrate was carefully removed with argon blowing. Non-coated materials were washed out by a stream of distilled water, and the substrate was gently dried by argon blowing. Finally, we repeated the washing step three times and stored the samples in a desiccator for analysis.

### 2.6. Characterization

The mesoporous structure of the aerogel was investigated by field-emission scanning electron microscopy (FE-SEM, Hitachi, Tokyo, Japan, SU8220) and transmission electron microscopy (TEM, JEOL, Tokyo, Japan, JEM-2100). The surface of the aerogel was analyzed by Fourier transform-infrared spectroscopy (FT-IR, Shimadzu, Kyoto, Japan, IRTracer-100) and X-ray photoelectron spectroscopy (XPS, Thermo Fisher Scientific, Waltham, MA, USA, ESCALAB 250XI). XPS spectra were recorded by Al Kα radiation (1486.6 eV) under ultrahigh vacuum (1.0 × 10^−10^ torr). The contact angle of a water droplet was measured using a drop shape analyzer (Krüss, Hamburg, Germany, DSA-100) to verify changes in hydrophobicity of the catechol-coated surfaces. Nitrogen adsorption–desorption measurements were carried out at −196 °C using a BELSORP-Mini II instrument. The samples were degassed at 120 °C for 24 h before the measurements. The specific surface area was calculated by the Brunauer–Emmett–Teller (BET) method. The thermal conductivity of the aerogel-impregnated glass wool sheet was measured using a modified transient plane source (MTPS) sensor in a thermal conductivity analyzer (C-Therm, Fredericton, NB, Canada, TCi). Statistical analysis was performed using one-way analysis of variance (ANOVA) followed by the Dunnett’s post hoc test. The probability value (*p*) < 0.05 was considered to indicate significant difference.

## 3. Results and Discussion

In general, APD-processed hydrophobic silica aerogel is produced by substituting the solvent in the wet gel with a hydrophobic solvent and then treating with a silylation agent such as TMCS. However, for surface modification with catechol derivatives, it is important to select a suitable exchanging solvent considering the strength of interaction with the modifier. Figure 1a is a schematic illustration of aerogel production through surface modification of catechol derivatives. First, we chose TBC and methanol as the surface modifier and solvent, respectively. The *tert*-butyl group of TBC induces hydrophobicity on the silica surfaces, similar to the methyl group of TMCS. Methanol dissolves the TBC molecules well and assists in their permeation into the gel network. In Figure 2, the morphologies of the silica aerogels are obtained via SEM according to the surface modification methods and then are compared. In the case of xerogel without surface modification, pores disappear due to shrinkage of the gel networks during the APD process (Figure 2a). However, the TBC- and TBC and HA-modified aerogels show the characteristics of type IV mesoporous materials in N_2_ physisorption isotherms (Appendix A) [26]. After TBC surface treatment, mesoporous silica networks with an average pore size of 23 nm are preserved with a specific surface area of 308 m^2^/g. In comparison, the average pore size of xerogel was 3.23 nm, and the surface area was 98 m^2^/g. (Figure 2b, Table 1). These results indicate that the *tert*-butyl group of TBC contributes to the surface hydrophobicity for a “spring-back” effect. Next, we add HA to the TBC solution with an expectation of two roles. One is the fabrication of a thin catechol coating layer. The catechol derivatives are easily oxidized and polymerized under basic conditions [27]. In these polymerization processes, the primary amine suppresses the polymerization of catechol molecules by Michael addition and Schiff base reactions [28,29,30]. By adding HA to the TBC coating solution, we can adjust the thickness of the coating layer and prevent the mesopore from clogging with coating materials. The second role of the addition of HA to the TBC solution is the enhancement of hydrophobicity owing to the alkyl chain exposed on the surface. As a result of the treatment of TBC and HA, the specific surface area and average pore diameter increase to 340 m^2^/g and 25 nm, respectively, compared to those of the TBC-modified aerogel (Figure 2c, Table 1).

The surface structure of the modified aerogels is analyzed by FT-IR spectroscopy (Figure 2e,f). The absorption peaks near 1635, 1070, 948, and 799 cm^−1^ appear commonly in all silica aerogel structures and represent the physically adsorbed water molecules deformation vibrations, Si-O-Si asymmetric stretching vibrations, Si-O in-plane stretching vibrations, and Si-O symmetric vibrations, respectively [31]. The TBC-containing aerogels show peaks at 2980, 2960, 1466, and 1380 cm^−1^, which are not seen in the xerogel (yellow and blue boxed region). These peaks indicate C-H asymmetric stretching, symmetric stretching, and deformation vibrations originating from the *tert*-butyl group in TBC molecules. In addition, we performed XPS analysis because it was difficult to find distinct changes between TBC and TBC + HA modified aerogels in FT-IR due to the small amount of the coating layer. In the N 1s XPS spectra, the TBC + HA aerogel showed a peak at 400.4 eV, which is indicating a C-N bond of secondary amine (Appendix A) [32]. This peak indicates the reaction between TBC and HA and is evidence of the two roles of HA: suppression of polymerization and enhancement of hydrophobicity. The control TMCS-modified aerogel exhibited the absorption peaks at 1258, 845, and 758 cm^−1^ are attributed to the symmetric deformation vibrations and stretching vibrations of Si-C bonds. In addition, it showed a specific surface area of 473 m^2^/g and pore diameters of ≈35 nm with the characteristic of type Ⅳ mesoporous materials in the N_2_ physisorption isotherm. (Figure 2d and Appendix A, Table 1).

TBC coating has shown potential for the hydrophobization of silica aerogel; however, it is known that catechol molecules have weak binding affinity to SiO_2_ surfaces and strong binding affinity to metal oxides (e.g., Al_2_O_3_, Fe_2_O_3_, and TiO_2_) surfaces (Figure 1b) [33,34,35]. To improve the adhesion to SiO_2_, we introduce catecholamine and dopamine, which can be universally coated on any surface [27,36]. However, another problem arises when only dopamine is used as a surface modifier. The mesoporous structure collapses similar to that of xerogel: by the uncontrollable polymerization of dopamine and its hydrophilic properties (Figure 3). This problem cannot be solved even when the coating time is reduced from 24 h to 10 min. To address this problem, we apply TBC and HA to the PDA coating. TBC and HA are expected to suppress the formation of PDA and impart hydrophobicity (Figure 1b). To verify this hypothesis, we obtain contact angle data for PDA, PDA + HA, PDA + TBC, and PDA + TBC + HA coatings on SiO_2_ wafers (Figure 4a). The contact angle of the PDA film is 48.2°, which indicates insufficient hydrophobicity for the “spring-back” effect. We confirm an increase in hydrophobicity as the contact angle of the PDA + HA and PDA + TBC films increase from 48.2° to 67.6° and 70.4°, respectively. However, in the case of PDA + HA, the mesoporous silica network is not maintained because of weak hydrophobization (Appendix A). On the other hand, we observe a mesoporous morphology in PDA + TBC modified aerogels, which has a specific surface area of 375 m^2^/g and pore diameters of ≈23 nm (Figure 4b, Table 1). The associated N_2_ adsorption–desorption isotherm and BJH pore size distribution graphs are shown in Appendix A. We infer that the *tert*-butyl group provides better hydrophobicity than HA and suppresses the polymerization of dopamine effectively. When TBC and HA are used simultaneously, the contact angle is the highest on the SiO_2_ wafer due to the synergetic effect, and the specific surface area of the aerogel increases slightly to 377 m^2^/g compared to that of the PDA + TBC aerogel (Figure 4c, Table 1). Polymerization suppression of PDA is also confirmed by observing the color of the catechol-modified aerogels (Appendix A). The color fade with an increasing ratio of TBC to PDA and with the addition of HA. This pale color indicates a thin catechol coating and the possibility of increased pore size by controlling catechol polymerization.

Aerogels are effective thermal insulators because aerogels are mainly composed of air in small nanopores. Before analyzing the thermal insulating property, we confirmed whether the hydrophobic surface treatment contributed to the preservation of the mesoporous structure through the “spring-back” effect through TEM analysis. As a result of the TEM analysis, in the case of xerogel, the mesopore was not preserved at all, but the samples treated with catechol modifiers showed mesoporous structures similar to TMCS-modified aerogels (Figure 5a–d). These results are direct evidence of the formation of a hydrophobic thin layer on the silica skeleton without clogging the mesopore. Therefore, the synthesized aerogel is impregnated into glass wool sheets for the fabrication of thermal insulation blankets; then, the thermal conductivity is measured and analyzed to see the efficiency of catechol modification. We then modify the aerogel-impregnated glass wool sheet with TBC + HA, PDA + TBC + HA, and TMCS (Appendix A). Thermal conductivities are measured using an MTPS sensor after placing a 1 kg weight on the fabricated glass wool sheets (Figure 5e). The xerogel has the highest thermal conductivity of 48.7 mWm^−1^K^−1^ because it only contains a small volume with air. The TBC + HA- and PDA + TBC + HA-modified samples show thermal conductivities of 37.9 and 40.4 mWm^−1^K^−1^, respectively, lower than that of the xerogel (Figure 5f). These values showed comparable performance to aerogel-impregnated blankets measured under the similar conditions in previously reported papers [37,38]. The thermal conductivity of the eco-friendly catechol-modified aerogel is higher than that of the TMCS-modified one but shows considerably higher heat insulation compared to that of xerogel. This economical water-glass aerogel suggests possible applications in the thermal insulation blanket industry.

## 4. Conclusions

In summary, we have developed a novel catechol-modified water-glass aerogel without the use of expensive and hazardous materials. Catechol derivatives are used instead of TMCS as surface modifiers. The *tert*-butyl group of TBC and the alkyl chain of HA impart hydrophobicity and suppress the polymerization of the catechol group. Dopamine is introduced to increase the binding strength between SiO_2_ and catechol. A combination of TBC, PDA, and HA is able to produce a water-glass aerogel similar to the TMCS-modified version. Finally, we impregnate the water-glass aerogel into glass wool sheets and modify them with catechol derivatives. This technique has commercial potential as a cost-effective thermal insulation blanket.

## Figures and Tables

**Figure 1 nanomaterials-10-02406-f001:**
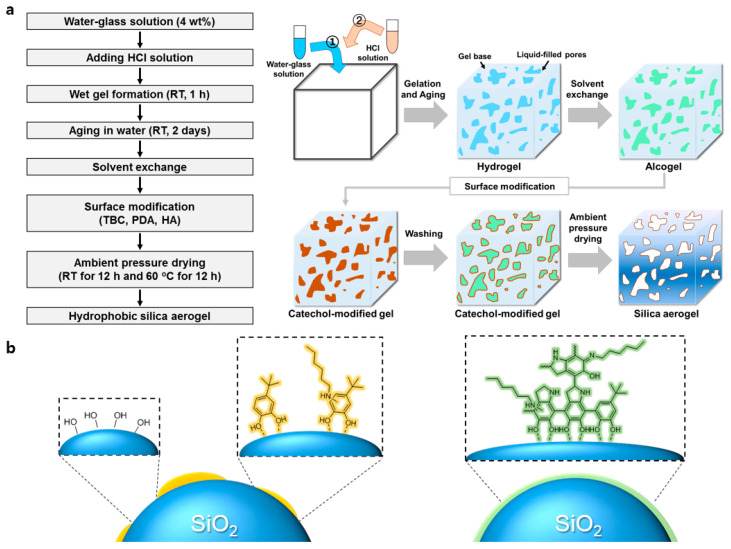
Schematic of (**a**) the synthesis process of catechol-modified water-glass-based silica aerogel, and (**b**) the proposed surface structure of 4-*tert*-butylcatechol (TBC) + hexylamine (HA) case (left) and polydopamine (PDA) + TBC + HA case (right).

**Figure 2 nanomaterials-10-02406-f002:**
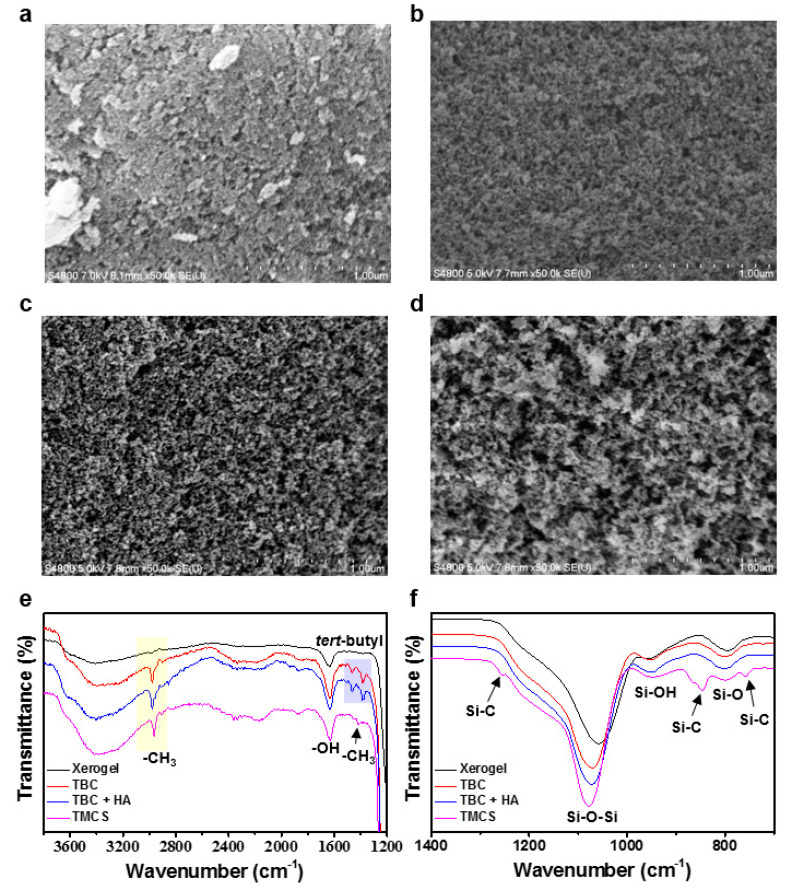
Scanning electron microscopy (SEM) images of (**a**) xerogel, (**b**) TBC, (**c**) TBC + HA, and (**d**) trimethylchlorosilane (TMCS)-modified aerogel. (**e**,**f**) Fourier transform-infrared spectroscopy (FT-IR) spectra of xerogel, TBC-modified aerogel, TBC + HA-modified aerogel, and TMCS-modified aerogel.

**Figure 3 nanomaterials-10-02406-f003:**
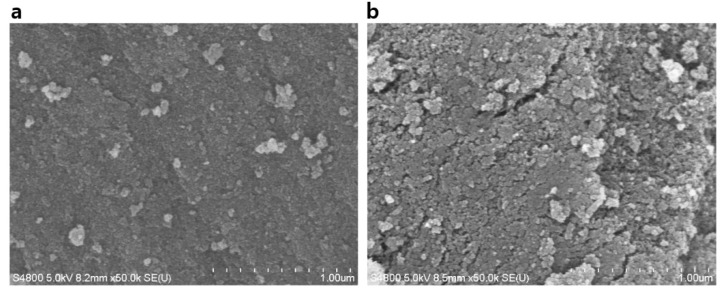
SEM images of PDA-modified aerogel incubated for (**a**) 10 min and (**b**) 24 h.

**Figure 4 nanomaterials-10-02406-f004:**
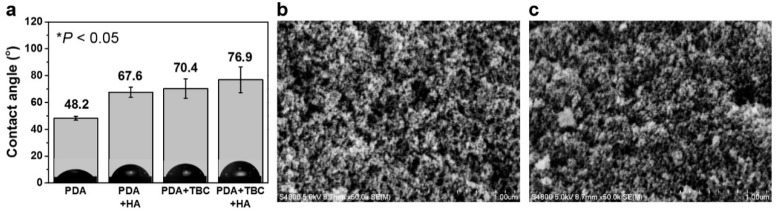
(**a**) Water contact angles on catechol-coated SiO_2_ wafers incubated for 12 h with PDA, PDA + HA, PDA + TBC, and PDA + TBC + HA. (*p* < 0.05, one-way ANOVA) SEM images of (**b**) PDA + TBC and (**c**) PDA + TBC + HA-modified aerogel.

**Figure 5 nanomaterials-10-02406-f005:**
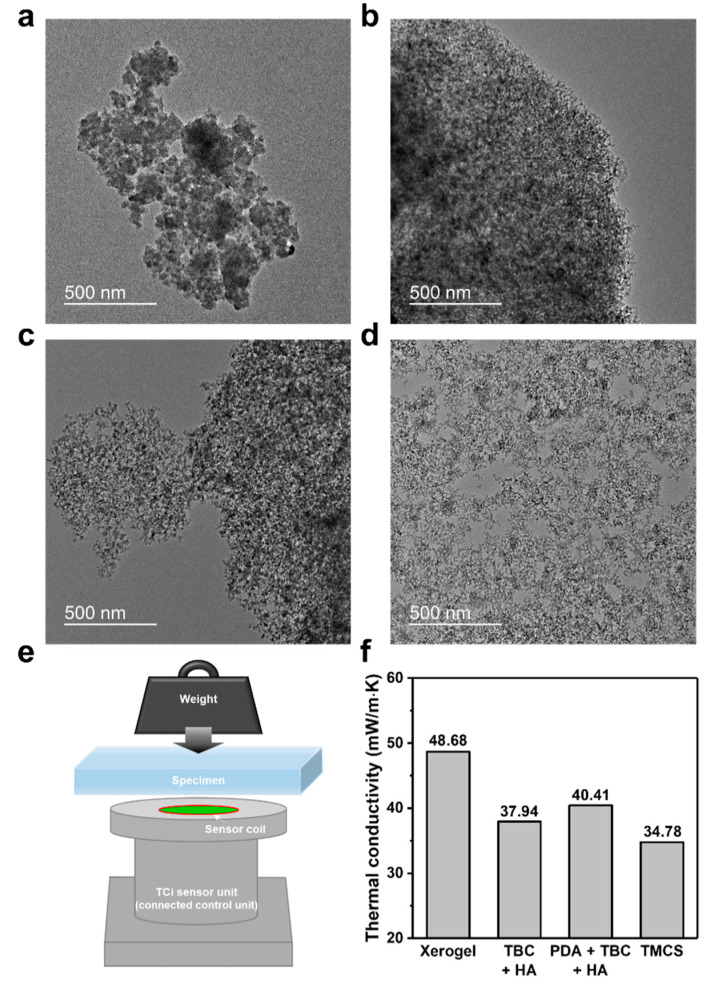
Transmission electron microscopy images of (**a**) xerogel, (**b**) TBC + HA, (**c**) PDA + TBC + HA, and (**d**) TMCS-modified silica aerogel (magnification of 13.5k). (**e**) Schematic of thermal conductivity analyzer using an modified transient plane source (MTPS) sensor (TCi, C-Therm). (**f**) Thermal conductivities on glass wool sheets impregnated by surface-modified silica aerogel.

**Table 1 nanomaterials-10-02406-t001:** Results of Brunauer–Emmett–Teller (BET) analysis.

Physical Property	Xerogel	TBC	TBC + HA	PDA + TBC	PDA + TBC + HA	TMCS
BET surface area (m^2^/g)	98	308	340	375	377	473
Pore volume (cm^3^/g)	0.08	1.77	2.13	2.14	1.93	4.15
Average pore diameter (nm)	3.23	22.90	25.02	22.82	20.52	35.11

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
