# Peer review of "Eco-Friendly Synthesis of Water-Glass-Based Silica Aerogels via Catechol-Based Modifier"

_nanomaterials, 2020, doi:10.3390/nano10122406_

Round 1

Reviewer 1 Report

The article entitled “Eco-friendly synthesis of silica nanoparticle-based aerogels via catechol-based modifier” presents the demonstration of catechol-based synthetic procedures towards fabricating eco-friendly aerogels through applying the good adhesion properties of catechol derivatives to 55 silica frameworks in order to form hydrophobic surfaces without toxic TMCS. Further, the fabricated aerogels, water-glass based silica wet gels, were compared with TMCS-modified aerogel and xerogel. Finally, the water-glass aerogel was impregnated into glass wool sheets and modified them with catechol derivatives. The authors demonstrated that the introduced dopamine could be able to produce strength between SiO2 and catechol. The study was well performed and executed. Thus, it could be publishable after minor revisions notified below.

The authors should state other reported eco-friendly strategies in the introduction.

 I would suggest taking TEM as the morphology and surface properties are important properties.

Statistical analysis in Figure 3D?

Author Response

General Comments: The article entitled “Eco-friendly synthesis of silica nanoparticle-based aerogels via catechol-based modifier” presents the demonstration of catechol-based synthetic procedures towards fabricating eco-friendly aerogels through applying the good adhesion properties of catechol derivatives to 55 silica frameworks in order to form hydrophobic surfaces without toxic TMCS. Further, the fabricated aerogels, water-glass based silica wet gels, were compared with TMCS-modified aerogel and xerogel. Finally, the water-glass aerogel was impregnated into glass wool sheets and modified them with catechol derivatives. The authors demonstrated that the introduced dopamine could be able to produce strength between SiO2 and catechol. The study was well performed and executed. Thus, it could be publishable after minor revisions notified below.

Response: We appreciate the reviewer for the prudent review on our manuscript and providing constructive comments. The comments were greatly helpful to improve the quality of our work. Below, we provide detailed responses and revised work.

Comment 1: The authors should state other reported eco-friendly strategies in the introduction.

Response: We appreciate the valuable comment by the reviewer. We tried to look into the literature regarding an eco-friendly surface treatment strategy for silica aerogels similar to our research, but appropriate examples that induces “spring-back” by using an environment-friendly material such as catechol in the hydrophobic surface treatment process of wet gel to mediate the ambient pressure drying process, were quite limited. As different strategies, there have been many studies on the organic material-based eco-friendly aerogel. So, we introduced eco-friendly aerogels with organic networks, and their drawbacks as well as the limited research of inorganic aerogels without silylation agents. We also justified the necessity of surface modified silica aerogel with catechol group which are relatively less toxic comparing to silylation agents.

The added description and references are as follows.

Therefore, the development of an inexpensive and eco-friendly surface treatment method is required, but researches on alternative of silylation agents on the surface of water glass aerogels are limited. As a different approach for eco-friendly aerogel, many studies have been conducted to develop the aerogels based on organic networks with low toxicity such as cellulose, lignin, and chitosan-based aerogels and increase their mechanical properties [19-23]. However, these materials still have a disadvantage of relatively low flame retardancy than silica aerogel, it is necessary to find a proper surface modifier imparting the hydrophobicity to the inorganic silica aerogel without the silylation agents.” (Page 2, line 46-53)

Revised references
19. Long, L.-Y.; Weng, Y.-X.; Wang, Y.-Z. Cellulose Aerogels: Synthesis, Applications, and Prospects. Polymers 2018, 10, 623.
20. Grishechko, L.I.; Amaral-Labat, G.; Szczurek, A.; Fierro, V.; Kuznetsov, B.N.; Pizzi, A.; Celzard, A.
New tannin–lignin aerogels. Ind Crop. Prod. 2013, 41, 347–355.
21. Kadib, A. E.; Bousmina, M. Chitosan Bio-Based Organic-Inorganic Hybrid Aerogel Microspheres. Chem. Eur. J. 2012, 18, 8264-8277.
22. Takeshita, S.; Yoda, S. Upscaled Preparation of Trimethylsilylated Chitosan Aerogel. Ind. Eng. Chem. Res. 2018, 57, 10421-10430.
23. Takeshita, S.; Yoda, S. Chitosan Aerogels: Transparent, Flexible Thermal Insulators. Chem. Mater. 2015, 27, 7569-7572.

Comment 2: I would suggest taking TEM as the morphology and surface properties are important properties.

Response: Thank you for the valuable comments. We conducted TEM analysis for four samples with the thermal conductivity results, shown in revised manuscript’s Figure 5f. TEM analysis indicate TBC+HA and PDA+TBC+HA samples with hydrophobic surface have mesoporous mesh structures similar to those of TMCS sample, whereas xerogel has no significant mesopores. These TEM results matched the BET analysis in table 1 in the revised manuscript and figure S1 in the revised supporting information. We added descriptions and data to Figure 5 in the revised manuscript as the reviewer suggested.

Before analyzing the thermal insulating property, we confirmed whether the hydrophobic surface treatment contributed to the preservation of the mesoporous structure through the “spring-back” effect through TEM analysis. As a result of the TEM analysis, in the case of xerogel, the mesopore was not preserved at all, but the samples treated with catechol-modifiers showed mesoporous structures similar to TMCS-modified aerogels (Figure 5). These results are direct evidence of formation of hydrophobic thin layer on the silica skeleton without clogging the mesopore.” (Page 7, line 226-231)

Figure 5. TEM images of (a) xerogel, (b) TBC+HA, (c) PDA+TBC+HA, and (d) TMCS-modified silica aerogel (magnification of 13.5k). (e) Schematic of thermal conductivity analyzer using an MTPS sensor (TCi, C-Therm). (f) Thermal conductivities on glass wool sheets impregnated by surface-modified silica aerogel.

Comment 3: Statistical analysis in Figure 3D?

Response: We appreciate your valuable feedback. As you said, we performed statistical analysis of the contact angle results in Figure 3D (Figure 4a in the revised manuscript). We performed the normality test and Levene’s test and confirmed the statistical significance through the one-way analysis of variance (ANOVA) followed by the Dunnett’s post hoc test (p < 0.05). Therefore, we have added p-value and ANOVA descriptions in the revised manuscript’s Figure 4a and section 2.6.

“Statistical analysis was performed using one-way analysis of variance (ANOVA) followed by the Dunnett’s post hoc test. The probability value (P) < 0.05 was considered to indicate significant difference.
(Page 3, line 134-136)

Figure 4. (a) Water contact angles on catechol-coated SiO2 wafers incubated for 12 h with PDA, PDA+HA, PDA+TBC, and PDA+TBC+HA. (P < 0.05, one-way ANOVA) SEM images of (b) PDA+TBC and (c) PDA+TBC+HA modified aerogel. (Page 7, line 217-219)

Reviewer 2 Report

The manuscript although well- written needs some improvisation.

1) The title needs to be specific and reflect the content. For eg., current title is "synthesis of silica nanoparticle-based aerogels via catechol-based modifier". But the author's work is on specific aerogel called water-glass silica. Please modify the title to reflect this.

2) In section Materials and methods, the entire 2.1-2.6 section must be reshuffled.

a)"The gelation process was completed within 1 h. Once the wet gel had formed, a small amount..". What is small amount. the authors must be specific. Also, the following sentence "the samples were aged.." What temperature and atmosphere was used for aging.

b). Section 2.3 How were the aerogels dried? Is it the same wet gel from section 2.2 that undergoes hydrophobization.

c) Section 2.4 Catechol modified aerogel. Is this the same wet gel with different modification? Is it the one the authors use in this study?

It is unclear to the reader why these many different treatments and at the end which aerogel the authors intend  to use. Also the drying method/procedure of obtaining these aerogels are not mentioned.

d) Section 2.5 - Why are the authors coating the Si Wafer?

e) Section 2.6 - must be combined with section 2.2

A nice step by step schematic on process would help under the reader better on material preparation.

3) Fig1. top part of the schematic must be more detailed.

4) Figure S2 must be moved to main text.

5) Fig 2a-d higher resolution images required to show the difference in particle sizes obtained.

6) Fig 3. a-c same SEM micrographs were used. Either replace it or remove it.

7)

Author Response

Recommendation: Publish after major revision.

General Comments: The manuscript although well- written needs some improvisation. It is unclear to the reader why these many different treatments and at the end which aerogel the authors intend to use. Also the drying method/procedure of obtaining these aerogels are not mentioned. A nice step by step schematic on process would help under the reader better on material preparation.

Response: We greatly appreciate the reviewer for the careful review on our manuscript and providing valuable comments. Below, we provide point-by-point responses to the reviewer’s comments.

Comment 1: The title needs to be specific and reflect the content. For eg., current title is "synthesis of silica nanoparticle-based aerogels via catechol-based modifier". But the author's work is on specific aerogel called water-glass silica. Please modify the title to reflect this.

Response: We appreciate the valuable comment by the reviewer. As the reviewer pointed out, the original title was revised as follows “Eco-friendly synthesis of water-glass-based silica aerogels via catechol-based modifier” (Page 1, line 2-3). In this manuscript, we showed the catechol-modified surface on the water-glass-based aerogel considering commercialization. However, it would be appreciated for you to be aware that this method is applicable to tetramethyl orthosilicate-based aerogel as well.

Comment 2: In section Materials and methods, the entire 2.1-2.6 section must be reshuffled.
Comment 2-a): "The gelation process was completed within 1 h. Once the wet gel had formed, a small amount..". What is small amount. the authors must be specific. Also, the following sentence "the samples were aged.." What temperature and atmosphere was used for aging.

Response: We thank the reviewer for providing valuable comments. As the reviewer commented, we added the specific information about the amount of water and aging conditions in the section 2.2. The previous descriptions were modified as the following sentences. Additional information is highlighted in blue.

All steps of wet-gel preparation were conducted under ambient temperature and atmosphere. First, 60 mL of the water-glass solution was mixed with 350 mL of deionized water. Next, 10 mL of diluted water-glass solution was mixed with 3.5 mL of 10 vol% HCl solution and poured into a mold. The gelation process was completed within 1 h. Once the wet gel had formed, we added 5 mL of deionized water and then aged at room temperature for 2 days to strengthen the gel network. After aging, the wet gel was pulverized into small pieces and washed 3 times with 20 mL of deionized water before further surface modification.” (Page 2, line 81-87)

Comment 2-b): Section 2.3 How were the aerogels dried? Is it the same wet gel from section 2.2 that undergoes hydrophobization.

Response: Thank you very much for your valuable comments. As pointed by the reviewer, we added more details on the drying method and wet-gel information used for hydrophobization in the section 2.3. After decanting the solvent, the surface-modified aerogels were dried for 12 h at room temperature and ambient pressure, and further dried in an oven at 60 oC for 12 h. We performed all surface treatments using the wet gel synthesized in section 2.2. The previous descriptions were modified as the following sentences. Additional information is highlighted in blue.

For hydrophobization with TMCS, the wet gels formed in section 2.2 were used. Firstly, the water in the wet gel was replaced with the isopropanol/hexane solution through several solvent exchange processes. After the solvent exchange process, the wet gels were chemically modified in 6.25 vol% TMCS/hexane solution for 24 h. The TMCS-modified wet gel was washed 3 times with hexane to remove residual modifiers and byproducts. After the solvent was decanted, the hydrophobized wet gel was dried for 12 h at room temperature and ambient pressure, and further dried in an oven at 60 oC for 12 h to obtain a TMCS-modified aerogel. The xerogel could be produced by the same drying process without surface modification.” (Page 3, line 95-102)

Comment 2-c): Section 2.4 Catechol modified aerogel. Is this the same wet gel with different modification? Is it the one the authors use in this study?

Response: We appreciate the reviewer’s valuable comments. As the reviewer pointed out, we used the same wet gel which was prepared in section 2.2 for various surface modifications. All catechol-modified aerogels treated in this study were synthesized by the method described in section 2.4. The previous explanations were modified as the following sentences. Additional information is highlighted in blue.

The wet gel prepared in section 2.2 was used for the surface modification applying catechol. The water in the wet gel was replaced with alcohol solvents such as methanol and ethanol through several solvent exchange processes. After the solvent exchange process, the wet gels were chemically modified in 10.5 mM catechol solution (TBC or a mixture of dopamine hydrochloride and TBC, 1:5 molar ratio) for 24 h. The 210 mM HA reagent was added to the catechol solution for hydrophobicity. The catechol-modified wet gel was washed 5 times with alcohol to remove residual modifiers and byproducts. Finally, the surface-modified aerogel was dried for 12 h at room temperature and ambient pressure, and further dried in an oven at 60 oC for 12 h.(Page 3, line 104-111)
Comment 2-d): Section 2.5 - Why are the authors coating the Si Wafer?

Response: Thank you very much for the reviewer’s critical comment. To demonstrate the properties of the catechol-coated surface, we coated various catechol molecules on the silicon wafer and compared the water contact angle. We selected the silicon wafer with a 200 nm oxide layer as a substrate because its components are similar to that of silica aerogel surfaces, and the smoothness of the surface is helpful for elaborating film analysis by water contact angle. We added more detailed information in the revised manuscript section 2.5. Additional information is highlighted in blue.

To demonstrate the properties of the catechol-coated surface, the coating on the silicon wafer was performed by a simple dip-coating method. We selected the silicon wafer with a 200 nm oxide layer as a substrate because its components are similar to that of silica aerogel surfaces, and the smoothness of the surface is helpful for elaborating film analysis. The coating solution contained 10.5 mM of dopamine hydrochloride (or a mixture of dopamine hydrochloride and TBC, 1:5 molar ratio) and 210 mM of HA in methanol (or ethanol). SiO2 substrates were immersed vertically in the coating solution for 12 h. The substrate was carefully removed with argon blowing. Non-coated materials were washed out by a stream of distilled water, and the substrate was gently dried by argon blowing. Finally, we repeated the washing step three times and stored the samples in a desiccator for analysis.” (Page 3, line 113-121)

Comment 2-e): Section 2.6 - must be combined with section 2.2

Response: We appreciate your valuable feedback. According to the suggestion, we combined section 2.2 and 2.6 with some modifications.

“2.2. Preparation of water-glass based silica wet gel and aerogel-impregnated glass wool sheet

All steps of wet-gel preparation were conducted under ambient temperature and atmosphere. First, 60 mL of the water-glass solution was mixed with 350 mL of deionized water. Next, 10 mL of diluted water-glass solution was mixed with 3.5 mL of 10 vol% HCl solution and poured into a mold. The gelation process was completed within 1 hour. Once the wet gel had formed, we added 5 mL of deionized water and then aged at room temperature for 2 days to strengthen the gel network. After aging, the wet gel was pulverized into small pieces and washed 3 times with 20 mL of deionized water before further surface modification. Aerogel-impregnated glass wool sheets were prepared by the following method. Prior to gelation and surface modification, glass wool sheets were cut to 4 cm × 6 cm × 1 cm cuboids. Subsequently, the glass wool sheets were soaked in 30 ml of diluted water-glass solution. Next, the gelation process was carried out by adding 10.5 mL of acidic HCl solution. Next the wet gel-glass wool composite was aged for 2 days. After aging, the composite was washed 3 times with deionized water before further surface modification. The hydrophobization and drying steps were performed in the same manner as the aerogel surface modification steps described below.(Page 2, line 80-93)

Comment 3: Fig1. top part of the schematic must be more detailed.

Response: We appreciate the valuable comment by the reviewer. We added the overall flowchart for the synthesis of silica aerogel and the corresponding schematic illustration in more detail to Figure 1a. (Page 4, line 138-139)

Figure 1. Schematic of (a) the synthesis process of catechol-modified water-glass-based silica aerogel

Comment 4: Figure S2 must be moved to main text.

Response: We thank the reviewer for the careful comment. As the reviewer suggested, we moved Figure S2 to the main text and modified the figure number. Due to the additional figure insertions, the numbering of

previous figures have increased by one after Figure 2. (Page 6, line 189-190)

Figure 3. SEM images of PDA-modified aerogel incubated for (a) 10 min and (b) 24 h.

Comment 5: Fig 2a-d higher resolution images required to show the difference in particle sizes obtained.

Response: We appreciate the valuable comment by the reviewer. We modified the resolution and arrangement of the images in Figure 2 to make it easier to obtain the difference in particle size. (Page 5, line 167-169)

Figure 2. SEM images of (a) xerogel, (b) TBC, (c) TBC+HA, and (d) TMCS-modified aerogel. (e, f) FT-IR spectra of xerogel, TBC-modified aerogel, TBC+HA-modified aerogel, and TMCS-modified aerogel.

Comment 6: Fig 3. a-c same SEM micrographs were used. Either replace it or remove it.

Response: We thank the reviewer for the careful comment about this matter. As the reviewer suggested, we removed Figure 3a because it was a duplicate of Figure S2 in supporting information. (As the reviewer pointed out in comment 4, Figure S2 has been moved to the main text.) Additionally, we replaced Figure b and c to different images, and Figure 4 rearranged in the order mentioned in the revised manuscript. Additionally, we performed statistical analysis of the contact angle results in Figure 3D (Figure 4a in the revised manuscript). We performed the normality test and Levene’s test and confirmed the statistical significance through the one-way analysis of variance (ANOVA) followed by the Dunnett’s post hoc test (p < 0.05). Therefore, we have
added p-value and ANOVA descriptions in the revised manuscript’s Figure 4a. (Page 7, line 217-219)

Figure 4. (a) Water contact angles on catechol-coated SiO2 wafers incubated for 12 h with PDA, PDA+HA, PDA+TBC, and PDA+TBC+HA. (P < 0.05, one-way ANOVA) SEM images of (b) PDA+TBC and (c) PDA+TBC+HA modified aerogel.

Reviewer 3 Report

The approach described in this study is interesting. However, the study is insufficiently scientifically consolidated and some points could be greatly improved. The characterization of xerogel could be more detailed (specific surface, pore volume, etc.). Moreover, SEM observations are really of poor quality and provide little information. A TEM or even AFM study could provide much more information. A study by SAXS would perhaps also be interesting to conduct on these materials. Regarding the measurements of contact angles and thermal conductivity, a comparison with data from the literature would be useful.

Author Response

â–ª Reviewer: 3

Recommendation: Publish after major revision.

General Comments: The approach described in this study is interesting. However, the study is insufficiently scientifically consolidated, and some points could be greatly improved.

Response: We appreciate the reviewer for the prudent review on our manuscript and providing constructive comments. The comments were greatly helpful to improve the quality of our work. Below, we provide detailed responses and revised work.

Comment 1: The characterization of xerogel could be more detailed (specific surface, pore volume, etc.).

Response: We appreciate the valuable comment by the reviewer. As you advised, we conducted the Brunauer-Emmett-Teller (BET) method to determine the effectiveness of the catechol surface modification to mediate the ambient pressure drying (APD) process. As a result, the BET specific surface area was 98 m2/g and the pore volume was 0.08 cm3/g. These are very low values compared to the catechol-modified samples, and it is the evidence showing the hydrophobization thorugh the catechol surface treatment.
We thought these results were important, so the analysis results of xerogel were added to the (a, b) of Figure S1 and Table 1 of the main text. Thanks again for the valuable comment.

In comparison, the average pore size of xerogel was 3.23 nm, and the surface area was 98 m2/g. (Figure 2b, Table 1). These results indicate that the tert-butyl group of TBC contributes to the surface hydrophobicity for a “spring-back” effect. (Page 4, line 154-157)

Figure S1. N2 adsorption-desorption isotherms (a) and the corresponding BJH pore size distributions (b) of xerogel

Table 1. Results of BET analysis

Physical property

Xerogel

TBC

TBC+HA

PDA

+TBC

PDA

+TBC+HA

TMCS

BET surface area (m2/g)

98

308

340

375

377

473

Pore volume (cm3/g)

0.08

1.77

2.13

2.14

1.93

4.15

Average pore diameter (nm)

3.23

22.90

25.02

22.82

20.52

35.11

Comment 2: Moreover, SEM observations are really of poor quality and provide little information. A TEM or even AFM study could provide much more information. A study by SAXS would perhaps also be interesting to conduct on these materials.

Response: We appreciate your valuable feedback. As you said, the SAXS analysis would be very helpful for knowing the detailed nanostructrual morphology of our catechoal-modified aerogel sample. Unfornuately, due to the limited time for the revision, we couldn’t characterize SAXS, but it will be very helpful to conduct SAXS analysis in the further study.
Also, as you mentioned, we recognized that the current SEM results are insufficient to explain the difference between the aerogel samples (xerogel, catechol-modified aerogel, and TMCS-modified aerogel). Therefore, we not only confirmed the efficiency of the catechol coating through comparison with the BET result of xerogel, but also observed that the mesoporous structure through TEM analysis was well preserved in the catechol-modified aerogel sample, unlike xerogel. Your advice was very helpful in supplementing our paper. We added descriptions and data to Figure 5 in the revised manuscript.

Before analyzing the thermal insulating property, we confirmed whether the hydrophobic surface treatment contributed to the preservation of the mesoporous structure through the “spring-back” effect through TEM analysis. As a result of the TEM analysis, in the case of xerogel, the mesopore was not preserved at all, but the samples treated with catechol-modifiers showed a mesoporous structure similar to TMCS-modified aerogels (Figure 5). These results are direct evidence of formation of hydrophobic thin layer on the silica skeleton without clogging the mesopore.(Page 7, line 226-231)

Figure 5. TEM images of (a) xerogel, (b) TBC+HA, (c) PDA+TBC+HA, and (d) TMCS-modified silica aerogel (magnification of 13.5k). (e) Schematic of thermal conductivity analyzer using an MTPS sensor (TCi, C-Therm). (f) Thermal conductivities on glass wool sheets impregnated by surface-modified silica aerogel.

Comment 3: Regarding the measurements of contact angles and thermal conductivity, a comparison with data from the literature would be useful.

Response: We thank the reviewer for providing valuable comment. Firstly, we compared the previously reported contact angle data with our results. Lee et al. proposed mussel-inspired polydopamine (PDA) coating and reported an average of 47o of water contact angle on various substrates [1]. In addition, Zhang et al. also showed a water contact angle of 44o on silicon wafer through PDA coating [2]. The results of these two references showed similar values to our result of 48.2o. To introduce hydrophobicity, we used 4-tert-butyl catechol (TBC) and hexlyamine (HA) and confirmed the increase in contact angle. Similarly, You et al. mixed hexanethiol with PDA to modulate the surface properties and showed a water contact angle of 65o on a silicon wafer [3]. This result is similar to 67.6o of our PDA+HA coating because primary amine and thiol groups can cause nucleophilic addition reaction with the catechol group. The hexyl group exposed to the outside after bonding via Michael addition or Schiff reaction provides a similar level of hydrophobicity to the surface. Finally, we were able to increase the water contact angle by 76.9o through the combination of PDA, TBC, and HA. Although this value is lower than the previously reported TMCS water contact angle of 100o or more, it can induce a spring-back effect as confirmed by BET, SEM, and TEM.
To compare the thermal conductivity, we investigated literatures on aerogel-impregnated blanket using the same measuring equipment (TCi, C-Therm) and method (modified transient plane source). Vaclavik et al. treated amorphous aerogel in a fabric balnket and measured the thermal conductivity according to the thickness of the blanket [4]. As the thickness of the blanket increased from 3.5 mm to 6.6 mm, the thermal conductivity increased from 26 mWm-1K-1 to 48.9 mWm-1K-1. Our blanket was 10 mm thick and exhibited a thermal conductivity of 37.94 mWm-1K-1 in the TBC+HA and 40.41 mWm-1K-1 in the PDA+TBC+HA , so it can be said that our blankets showed better properties than those in this paper. In the same group, an amorphous aerogel with pore size of 20 nm was impregnated into a fabric blanket and the thermal conductivity was measured in 2017 [5]. The 11.12 nm thick blanket showed a thermal conductivity of 39.6 mWm-1K-1, which is similar to our experimental results. Therefore, our catechol-modified aerogel has potential applications in the thermal insulation blanket industry.The added description and references are as follows.

These values showed comparable performance to aerogel-impregnated blankets measured under the similar conditions in previously reported papers [37, 38].(Page 8, line 239-240)

References
1. Lee, H.; Dellatore, S.M.; Miller, W.M.; Messersmith, P.B. Mussel-inspired surface chemistry for multifunctional coatings. Science 2007, 318, 426-430.
2. Zhang, C.; Ma, M.-Q.; Chen, T.-T.; Zhang, H.; Hu, D.-F.; Wu, B.-H.; Ji, J.; Xu, Z.-K. Dopamine-triggered one-step polymerization and codeposition of acrylate monomers for functional coatings. ACS Appl. Mater. Interfaces 2017, 9, 34356-34366.
3. You, I.; Jeon, H.; Lee, K.; Do, M.; Seo, Y.C.; Lee, H.A.; Lee, H. Polydopamine coating in organic solvent for material-independent immobilization of water-insoluble molecules and avoidance of substrate hydrolysis. J. Ind. Eng. Chem. 2017, 46, 379-385.
4. Venkataraman, M.; Mishra, R.; Wiener, J.; Militky, J.; Kotresh, T.M.; Vaclavik, M. Novel techniques to analyse thermal performance of aerogel-treated blankets under extreme temperatures. J. Text. Inst. 2015, 106, 736-747.
5. Venkataraman, M.; Mishra, R.; Militky, J. Comparative analysis of high performance thermal insulation materials. J. Text. Eng. Fash. Technol. 2017, 2, 401-409.

Round 2

Reviewer 2 Report

The authors have responded to the comments and made necessary changes to the manuscript.

Reviewer 3 Report

Most of my comments have been taken into account by the authors. The paper may be published as is.